# CONTINUOUS CONVOLUTIONAL NEURAL NETWORKS FOR IMAGE CLASSIFICATION

## ABSTRACT

This paper introduces the concept of continuous convolution to neural networks and deep learning applications in general. Rather than directly using discretized information, input data is first projected into a high-dimensional Reproducing Kernel Hilbert Space (RKHS), where it can be modeled as a continuous function using a series of kernel bases. We then proceed to derive a closed-form solution to the continuous convolution operation between two arbitrary functions operating in different RKHS. Within this framework, convolutional filters also take the form of continuous functions, and the training procedure involves learning the RKHS to which each of these filters is projected, alongside their weight parameters. This results in much more expressive filters, that do not require spatial discretization and benefit from properties such as adaptive support and non-stationarity. Experiments on image classification are performed, using classical datasets, with results indicating that the proposed continuous convolutional neural network is able to achieve competitive accuracy rates with far fewer parameters and a faster convergence rate.

## 1 INTRODUCTION

In recent years, convolutional neural networks (CNNs) have become widely popular as a deep learning tool for addressing various sorts of problems, most predominantly in computer vision, such as image classification (Krizhevsky et al., 2012; He et al., 2016), object detection (Ren et al., 2015) and semantic segmentation (Ghiasi & Fowlkes, 2016). The introduction of convolutional filters produces many desirable effects, including: translational invariance, since the same patterns are detected in the entire image; spatial connectivity, as neighboring information is taken into consideration during the convolution process; and shared weights, which results in significantly fewer training parameters and smaller memory footprint.

Even though the convolution operation is continuous in nature (Hirschman & Widder, 1955), a common assumption in most computational tasks is data discretization, since that is usually how information is obtained and stored: 2D images are divided into pixels, 3D point clouds are divided into voxels, and so forth. Because of that, the exact convolution formulation is often substituted by a discrete approximation (Damelin & Miller, 2011), calculated by sliding the filter over the input data and calculating the dot product of overlapping areas. While much simpler to compute, it requires substantially more computational power, especially for larger datasets and filter sizes (Pavel & David, 2013). The fast Fourier transform has been shown to significantly increase performance in convolutional neural network calculations (Highlander & Rodriguez, 2015; Rippel et al., 2015), however these improvements are mostly circumstantial, with the added cost of performing such transforms, and do not address memory requirements.

To the best of our knowledge, all versions of CNNs currently available in the literature use this discrete approximation to convolution, as a way to simplify calculations at the expense of a potentially more descriptive model. In Liu et al. (2015) a sparse network was used to dramatically decrease computational times by exploiting redundancies, and in Graham (2014) spatial sparseness was exploited to achieve state-of-the-art results in various image classification datasets. Similarly, Riegler et al. (2016) used octrees to efficiently partition the space during convolution, thus focusing memory allocation and computation to denser regions. A quantized version was proposed in Wu et al. (2016) to improve performance on mobile devices, with simultaneous computational acceleration

and model compression. A lookup-based network is described in Bagherinezhad et al. (2016), that encodes convolution as a series of lookups to a dictionary that is trained to cover the observed weight space.

This paper takes a different approach and introduces the concept of continuous convolution to neural networks and deep learning applications in general. This is achieved by projecting information into a Reproducing Kernel Hilbert Space (RKHS) (Schölkopf & Smola, 2001), in which point evaluation takes the form of a continuous linear functional. We employ the Hilbert Maps framework, initially described in Ramos & Ott (2015), to reconstruct discrete input data as a continuous function, based on the methodology proposed in Guizilini & Ramos (2016). Within this framework, we derive a closed-form solution to the continuous convolution between two functions that takes place directly in this high-dimensional space, where arbitrarily complex patterns are represented using a series of simple kernels, that can be efficiently convolved to produce a third RKHS modeling activation values. Optimizing this neural network involves learning not only weight parameters, but also the RKHS that defines each convolutional filter, which results is much more descriptive feature maps that can be used for both discriminative and generative tasks. The use of high-dimensional projection, including infinite-layer neural networks Hazan & Jaakkola (2015); Globerson & Livni (2016), has been extensively studied in recent times, as a way to combine kernel-based learning with deep learning applications. Note that, while works such as Mairal et al. (2014) and Mairal (2016) take a similar approach of projecting input data into a RKHS, using the kernel trick, it still relies on discretized image patches, whereas ours operates solely on data already projected to these high-dimensional spaces. Also, in these works extra kernel parameters are predetermined and remain fixed during the training process, while ours jointly learns these parameters alongside traditional weight values, thus increasing the degrees of freedom in the resulting feature maps.

The proposed technique, entitled Continuous Convolutional Neural Networks (CCNNs), was evaluated in an image classification context, using standard computer vision benchmarks, and achieved competitive accuracy results with substantially smaller network sizes. We also demonstrate its applicability to unsupervised learning, by describing a convolutional auto-encoder that is able to produce latent feature representations in the form of continuous functions, which are then used as initial filters for classification using labeled data.

## 2 HILBERT MAPS

The Hilbert Maps (HM) framework, initially proposed in Ramos & Ott (2015), approximates real-world complexity by projecting discrete input data into a continuous Reproducing Kernel Hilbert Space (RKHS), where calculations take place. Since its introduction, it has been primarily used as a classification tool for occupancy mapping (Guizilini & Ramos, 2016; Doherty et al., 2016; Senanayake et al., 2016) and more recently terrain modeling (Guizilini & Ramos, 2017). This section provides an overview of its fundamental concepts, before moving on to a description of the feature vector used in this work, and finally we show how model weights and kernel parameters can be jointly learned to produce more flexible representations.

### 2.1 OVERVIEW

We start by assuming a dataset $\mathcal{D} = (\mathbf{x}, y)_{i=1}^{N}$, in which $\mathbf{x}_i \in \mathcal{R}^D$ are observed points and $y_i = \{-1, +1\}$ are their corresponding occupancy values (i.e. the probability of that particular point being occupied or not). This dataset is used to learn a discriminative model $p(y|\mathbf{x}, \mathbf{w})$, parametrized by a weight vector $\mathbf{w}$. Since calculations will be performed in a high-dimensional space, a simple linear classifier is almost always adequate to model even highly complex functions (Komarek, 2004). Here we use a Logistic Regression (LR) classifier (Freedman, 2005), due to its computational speed and direct extension to online learning. The probability of occupancy for a query point $\mathbf{x}_*$ is given by:

$$p(y_* = 1 | \Phi(\mathbf{x}_*), \mathbf{w}) = \frac{1}{1 + \exp\left(-\mathbf{w}^T \Phi(\mathbf{x}_*)\right)}, \tag{1}$$

where $\Phi(.)$ is the feature vector, that projects input data into the RKHS. To optimize the weight parameters $\mathbf{w}$ based on the information contained in $\mathcal{D}$, we minimize the following negative log-

likelihood cost function:

$$\mathcal{L}_{NLL}(\mathbf{w}) = \sum_{i=1}^{N} \left( 1 + \exp\left( -y_i \mathbf{w}^T \Phi(\mathbf{x}_i) \right) \right) + R(\mathbf{w}), \tag{2}$$

with $R(\mathbf{w})$ serving as a regularization term, used to avoid over-fitting. Once training is complete, the resulting model can be used to query the occupancy state of any input point $\mathbf{x}_*$ using Equation 1, at arbitrary resolutions and without the need of space discretization.

## 2.2 LOCALIZED LENGTH-SCALES

The choice of feature vector $\Phi(.)$ is very important, since it defines how input points will be represented when projected to the RKHS, and can be used to approximate popular kernels such that $k(\mathbf{x}, \mathbf{x}') \approx \Phi(\mathbf{x})^T \Phi(\mathbf{x}')$. In Guizilini & Ramos (2016), the authors propose a feature vector that places a set $\mathcal{M}$ of inducing points throughout the input space, either as a grid-like structure or by clustering $\mathcal{D}$. Inducing points are commonly used in machine learning tasks Snelson & Ghahramani (2006) to project a large number of data points into a smaller subset, and here they serve to correlate input data based on a kernel function, here chosen to be a Gaussian distribution with automatic relevance determination:

$$k(\mathbf{x}, \boldsymbol{\mu}, \Sigma) = \frac{1}{(2\pi)^{\frac{D}{2}} |\Sigma|^{\frac{1}{2}}} \exp\left( -\frac{1}{2} (\mathbf{x} - \boldsymbol{\mu})^T \Sigma^{-1} (\mathbf{x} - \boldsymbol{\mu}) \right), \tag{3}$$

where $\boldsymbol{\mu} \in \mathcal{R}^D$ is a location in the input space and $\Sigma$ is a symmetric positive-definite covariance matrix that models length-scale. Each inducing point maintains its own mean and covariance parameters, so that $\mathcal{M} = \{\boldsymbol{\mu}, \Sigma\}_{i=1}^M$. The resulting feature vector $\Phi(\mathbf{x}, \mathcal{M})$ is given by the concatenation of all kernel values calculated for $\mathbf{x}$ in relation to $\mathcal{M}$:

$$\Phi(\mathbf{x}, \mathcal{M}) = \left[ \begin{array}{cccc} k(\mathbf{x}, \mathcal{M}_1) & , & k(\mathbf{x}, \mathcal{M}_2) & , & \ldots & , & k(\mathbf{x}, \mathcal{M}_M) \end{array} \right]^T. \tag{4}$$

Note that Equation 4 is similar to the sparse random feature vector proposed in the original Hilbert Maps paper (Ramos & Ott, 2015), but with different length-scale matrices for each inducing point. This modification naturally embeds non-stationarity into the feature vector, since different areas of the input space are governed by their own subset of inducing points, with varying properties. To increase efficiency, only a subset of nearest neighbors can be used for feature vector calculation, while all others are set to zero. Indeed, this feature vector has been successfully applied to accurately reconstruct large-scale 3D datasets at a fraction of the computational cost required by other similar kernel-based techniques (Callaghan & Ramos, 2012), which makes it attractive for big data processing.

## 2.3 JOINT KERNEL LEARNING

In the original implementation, the parameters $\{\boldsymbol{\mu}, \Sigma\}_i$ of each kernel in $\mathcal{M}$ are fixed and calculated based on statistical information obtained directly from $\mathcal{D}$. Only the classifier weights $\mathbf{w}$ are optimized during the training process, according to Equation 2. However, this approach severely limits the descriptiveness of each kernel, especially if training data is not readily available for pre-processing. Here we show how the HM training methodology can be reformulated to include the optimization of all its parameters $\mathcal{P} = \{\boldsymbol{\mu}, \Sigma, \mathbf{w}\}_{i=1}^M$.

The key insight is realizing that the HM framework is analogous to a neural network layer (Haykin, 1998), in which input data is described as a Gram Matrix (Hazewinkel, 2001) in relation to the inducing set $\mathcal{M}$, such that:

$$\mathbf{h} = \sigma \left( K_{X\mathcal{M}} \cdot \mathbf{w} + b_h \right), \tag{5}$$

where $X = \{\mathbf{x}\}_{i=1}^N$ are input points, $K_{X\mathcal{M}}$ is a $N \times M$ matrix with rows $K_i = \Phi(\mathbf{x}_i, \mathcal{M})$ (i.e. with coefficients $K_{ij} = k(\mathbf{x}_i, \mathcal{M}_j)$) as defined by Equation 3, $\sigma$ is the sigmoid activation function and $b_h$ is an optional bias term. Within this alternative framework, standard back-propagation (LeCun et al.,

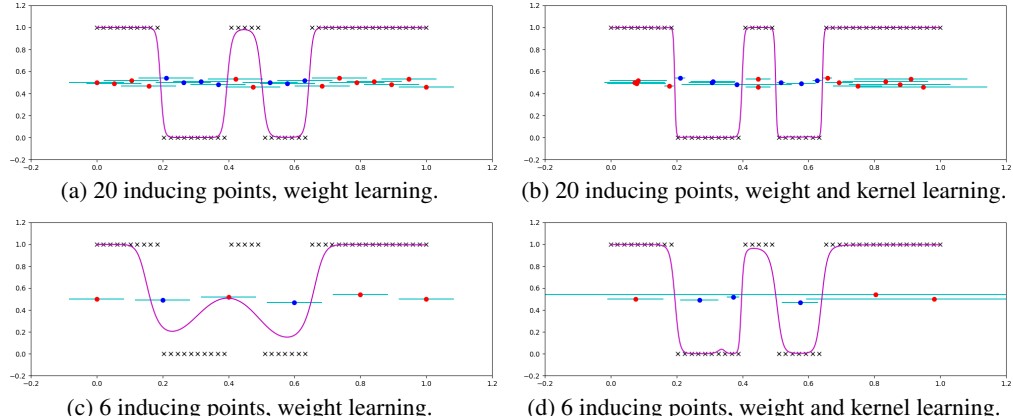

(a) 20 inducing points, weight learning.

(b) 20 inducing points, weight and kernel learning.

(c) 6 inducing points, weight learning.

(d) 6 inducing points, weight and kernel learning.

Figure 1: 1D example of the joint learning of weights and kernels parameters. Crosses indicate data points, dots indicate cluster centers (red for positive weights and blue for negative), horizontal lines indicate length-scales and the magenta line indicates optimal occupancy probabilities. The cluster centers are vertically offset purely to facilitate length-scale visualization.

1998) can be used to jointly optimize kernel parameters, using the corresponding partial derivatives:

$$\frac{\partial k(\mathbf{x}, \mathcal{M}_i)}{\partial \boldsymbol{\mu}_i} = -\frac{\mathbf{d}_i \Sigma_i^{-\frac{1}{2}}}{\sqrt{(2\pi)^D |\Sigma_i|}} \exp\left(-\frac{1}{2}\mathbf{d}_i^T \Sigma_i^{-\frac{1}{2}} \mathbf{d}_i\right) = \mathbf{d}_i \Sigma_i^{-\frac{1}{2}} \cdot k(\mathbf{x}, \mathcal{M}_i) \tag{6}$$

$$\frac{\partial k(\mathbf{x}, \mathcal{M}_i)}{\partial \Sigma_i} = \frac{\mathbf{d}_i^T \Sigma_i^{-\frac{1}{3}} \mathbf{d}_i - \frac{1}{|\Sigma_i|}}{\sqrt{(2\pi)^D |\Sigma_i|}} \exp\left(-\frac{1}{2}\mathbf{d}_i^T \Sigma_i^{-\frac{1}{2}} \mathbf{d}_i\right) = \left(\mathbf{d}_i^T \Sigma^{-\frac{1}{3}} \mathbf{d}_i - \frac{1}{|\Sigma_i|}\right) \cdot k(\mathbf{x}, \mathcal{M}_i), \tag{7}$$

which can be efficiently calculated during feature vector generation. An example of this joint learning process can be found in Figure 1, for a simple 1D classification problem. In the left column, the standard HM framework was used, with only weight learning, whereas in the right column kernel parameters were also learned, using the proposed HL framework (Hilbert Layer). In the top row 20 inducing points were used, initially equally spaced, while in the bottom row only 6 inducing points were used. Note that, for higher densities, HM converges to good results, however it fails to capture some occupancy behaviors in lower densities, due to its smaller descriptive power. On the other hand, HL is able to achieve considerably better results in both cases, with reasonable convergence even in lower densities. Particularly, in Figure 1b we can see how inducing points migrate to discontinuous areas, thus ensuring sharper transitions, and in Figure 1d one inducing points assumed a larger length-scale to reach a particular occupied area that was under-represented.

Lastly, note that, while the standard HM framework as described in Section 2.1 only addresses classification tasks, the proposed joint learning methodology can be easily modified to address general regression tasks, simply by removing the activation function $\sigma$ and optimizing a different loss function (i.e. mean squared error instead of cross-entropy).

# 3 CONTINUOUS CONVOLUTIONAL LAYER

In this section we show how the Hilbert layer, as defined by Equation 5, can be extended to a convolutional scenario, in which two functions in different RKHS are convolved to produce a third RKHS defining functions that approximate activation values. We start by formulating a closed-form solution to the continuous convolution between kernels that describe a feature vector for projection in the HL framework, move on to a formal definition of the Convolutional Hilbert Layer (CHL), and lastly describe how this novel framework can be used for image classification tasks.

### 3.1 Closed-Form Kernel Convolution

Convolution is a mathematical operation that takes two functions $f$ and $g$ and produces a third function $h = f * g$, that can be viewed as the amount of overlapping between both functions as one is reversed and shifted over the other. Formally, it is defined as:

$$h(t) = (f * g)(t) = \int_{-\infty}^{+\infty} f(t - \tau)g(\tau)d\tau. \tag{8}$$

Solving Equation 8 analytically can be a difficult (or even impossible) task for most functions that define real-world phenomena, due to their complexity. However, as shown in Sec. 2.2, the Hilbert Maps framework is able to approximate arbitrarily complex functions using simple kernels, by projecting input data into a high-dimensional RKHS. Although the proposed methodology can be applied to any two kernel functions with a closed-form convolution formula, for notation simplicity we assume, without loss of generality, that the kernels describing both functions are given by Equation 3. This choice greatly simplifies the problem because the convolution of two Gaussian distributions is also a Gaussian distribution with automatic relevance determination (Bromiley, 2003):

$$k(\mathbf{x}, \mathcal{M}_i) * k(\mathbf{x}, \mathcal{M}_j) = k(\mathbf{x}, \mathcal{M}_i + \mathcal{M}_j), \tag{9}$$

where $\mathcal{M}_i + \mathcal{M}_j = \{\boldsymbol{\mu}_i + \boldsymbol{\mu}_j, \Sigma_i + \Sigma_j\}$. Note that this new kernel does not model function states in the RKHS, but rather activation values, representing convolution results between $\mathcal{M}_i$ and $\mathcal{M}_j$, and can be queried at arbitrary resolutions using the HL framework described in Section 2.3. More importantly, it can be optimized using the same training methodology, to produce better generative or discriminative models.

### 3.2 Convolutional Hilbert Layer

Now that convolution between kernels that define functions in the RKHS has been established, here we show how convolution between two parameter sets $\mathcal{P}_f$ and $\mathcal{P}_g$, each representing a different function in its respective RKHS, is performed. Comparing to the Hilbert layer defined in Equation 5, the convolutional Hilbert Layer takes the form:

$$\mathbf{h} = \sigma \left( \sum_{fg} \left( \mathbf{w}_f \cdot \begin{bmatrix} \left[ K_{fg}^1 \right] \\ \left[ K_{fg}^2 \right] \\ \vdots \\ \left[ K_{fg}^{M_h} \right] \end{bmatrix} \cdot \mathbf{w}_g \right) + b_h \right) = \sigma \left( \sum_{fg} \left( \mathbf{w}_f \cdot \mathbf{K}_{fg} \cdot \mathbf{w}_g \right) + b_h \right) \tag{10}$$

where $K_{fg}^k$ is a $M_f \times M_g$ matrix, with coefficients $K_{fg} = k(\boldsymbol{\mu}_h^k, \mathcal{M}_f^i + \mathcal{M}_g^j)$. Note that $\mathbf{K}_{fg}$ is a block-matrix, so weight multiplications are performed independently for each entry before summation, which benefits the use of parallel computing for faster calculations. The output $\mathbf{h}$ are the weights that approximate $f * g$ in the RKHS defined by the cluster set $\mathcal{M}_h$, and together these compose the parameter set $\mathcal{P}_h = \{\mathcal{M}_h, \mathbf{h}\} = \{\boldsymbol{\mu}_h, \Sigma_h, \mathbf{w}_h\}$. Pseudo-code for this convolution process is given in Algorithm 1, where we can see how it operates: for each inducing point in $\mathcal{P}_f$ and $\mathcal{P}_g$,

---

**Algorithm 1** Convolutional Hilbert Layer

**Require:** Input $\mathcal{P}_f$ and filter $\mathcal{P}_g$ parameter sets, convolved $\mathcal{M}_h$ cluster set
**Ensure:** Convolved $\mathcal{P}_h$ parameter set
1: $\mathcal{P}_h \leftarrow \mathcal{M}_h$ % Convolved parameter set is initialized with cluster set values
2: $\mathbf{w}_h \leftarrow \mathbf{0}$     % Convolved parameter set weights are set to zero
3: **for** $\mathcal{P}_f^i \in \mathcal{P}_f$ **do**
4:     **for** $\mathcal{P}_g^j \in \mathcal{P}_g$ **do**
5:         **for** $\mathcal{P}_h^k \in \mathcal{P}_h$ **do**
6:             $w_h^k \mathrel{+}= w_f^i w_g^j \cdot k(\boldsymbol{\mu}_h^k, \mathcal{M}_i^f + \mathcal{M}_j^g)$
7:         **end for**
8:     **end for**
9: **end for**

---

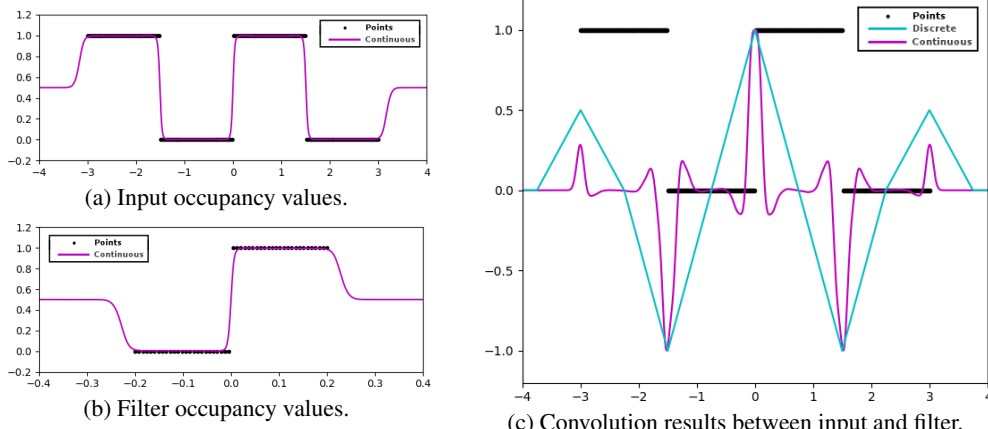

Figure 2: 1D example of continuous convolution between two occupancy functions. The input in (a) is convolved with the filter in (b) to produce the convolution results depicted in (c).

their kernel convolution is calculated (Equation 9) and used to evaluate all points in $\mathcal{P}_h$ (Eq. 3), each one contributing to its corresponding weight parameter $\mathbf{w}_h^k$. Multiple $P$ input channels and $Q$ filters can be incorporated by concatenating the various $\mathbf{K}_{fg}^{pq}$ into a single block-matrix, while augmenting $\mathbf{w}_f$ and $\mathbf{w}_g$ accordingly, such that:

$$H = \sigma \left( \sum_{fgp} \left( \boldsymbol{W}_f \cdot \boldsymbol{\mathcal{K}}_{fg} \cdot \boldsymbol{W}_g \right) + \mathbf{b}_h \right), \tag{11}$$

where $\mathcal{K}_{fg}$ is a $P \times Q$ block-matrix with entries $\mathcal{K}_{ij} = \mathbf{K}_{fg}^{pq}$, $\mathbf{W}_f = \left[ \left[ \mathbf{w}_f^1 \right], ..., \left[ \mathbf{w}_f^P \right] \right]$, $\mathbf{W}_g = \left[ \left[ \mathbf{w}_g^1 \right], ..., \left[ \mathbf{w}_g^Q \right] \right]$, $\mathbf{b}_h$ is now a $Q \times 1$ bias vector and $\mathcal{P}_h = \{\mathcal{M}_h, H\} = \{\boldsymbol{\mu}_h, \Sigma_h, \{\mathbf{w}\}_{q=1}^Q\}$ contains multiple weight channels defined in the same RKHS. An example of the output produced by a convolutional Hilbert Layer, for a simple 1D classification problem, is depicted in Figure 2, alongside the corresponding discrete convolution results. As expected, areas in the input that are similar to the filter have higher activation values, and these vary continuously throughout the input space, being able to capture small variations and partial matches to a higher detail than discrete convolution.

Lastly, note that pooling (Krizhevsky et al., 2012) can be naturally incorporated into the convolutional Hilbert layer simply by decreasing the density of clusters in the output RKHS. This process has two effects: 1) decrease computational cost, which allows for larger filter sizes and number of channels while combating over-fitting; and 2) decrease the resolution of the approximated continuous function, thus aggregating statistics of spatially close regions to capture patterns in a larger scale. An un-pooling effect is similarly straightforward, generated by increasing the density of clusters in the output RKHS, thus increasing the resolution of the approximated continuous function at the expense of a larger number of parameters.

### 3.3 Classification Topology

A diagram depicting how the proposed convolutional Hilbert layer can be applied to an image classification task, to create a Continuous Convolutional Neural Network (CCNN), is shown in Figure 3. The original image, composed of discrete data $\mathcal{D}^n = \{\mathbf{x}, \mathbf{y}^n\}$, with $\mathbf{x} \in \mathcal{R}^2$ being the pixel coordinates and $\mathbf{y}^n = [0, 1]^{C_0}$ their corresponding intensity values ($C_0$ is the number of input channels, i.e. 1 for grayscale and 3 for color images), is first modeled as a continuous function via projection to a RKHS. Note that the same RKHS is used to model all input images, defined by the cluster set $\mathcal{M}_{f0}$, and within this projection each image is represented with a different set of model weights $\mathbf{W}_{f0}^n$. The resulting parameter set $\mathcal{P}_f^n$ is convolved with the filters contained in $\mathcal{P}_{g1}$ to produce the hidden feature maps $\mathcal{P}_{f1}$, that also share the same RKHS for all input images, defined by $\mathcal{M}_{f1}$, but with individual model weights $\mathbf{W}_{f1}^n$. This process is repeated for each convolutional Hilbert layer,

with the final model weights being flattened to serve as input for a standard fully connected neural network, that performs the classification between different categories.

The convolutional trainable parameters to be optimized in this topology are the various cluster sets $\mathcal{M}_f = \{\boldsymbol{\mu}, \Sigma\}$, that define the RKHS for each feature map, and the various filter sets $\mathcal{P}_g = \{\boldsymbol{\mu}, \Sigma, \mathbf{W}\}$, that perform the transformation between feature maps in each layer. Note that each of these parameters represents a different property of the convolution process: $\boldsymbol{\mu}$ defines location, $\Sigma$ defines length-scale and $\mathbf{W}$ defines weight, and therefore should be treated accordingly during optimization. To guarantee positive-definitiveness in variance values, we are in fact learning a lower triangular matrix $V$, such that $\Sigma = V^T V$, which is assumed to be invertible (i.e. its determinant is not zero, or none of its main diagonal values are zero). While this property cannot be strictly guaranteed during the optimization process, in practice the noisy nature of stochastic gradient descent naturally avoids exact values of zero for trainable parameters, and at no point during experiments this assumption was broken.

To improve parameter initialization, we employ a continuous fully-convolutional auto-encoder (CCAE), that first encodes data into a lower-dimensional latent feature vector representation and then decodes it back to produce a reconstruction of the original input. Particularly, the encoding pipeline is composed by the convolutional Hilbert layers from the classification topology, and the decoding pipeline has these same layers in reverse order (without parameter sharing), as depicted in Figure 4. A lower-dimensional representation is achieved by decreasing the number of clusters used for feature map projection in deeper layers, thus simulating a pooling effect. Similarly, the number of clusters used for filter projection can also be modified, emulating different kernel sizes in standard discrete convolution, however since the location of these clusters is a trainable parameter their support is inherently adaptive, only changing in complexity as more clusters are added. In all experiments, inducing points were initialized with mean values equally spaced in the 2D space and with the same variance value, so that the distance between mean values is equal to two standard deviations (weight values were initialized randomly, using a truncated Gaussian distribution with mean 0 and variance 0.1).

## 4 EXPERIMENTAL RESULTS

Here we present and discuss experimental results performed to validate the proposed convolutional Hilbert layer in an image classification scenario[1]. Four different standard benchmarks were considered: the MNIST dataset, composed of 60000 training and 10000 test images with dimensionality $28 \times 28 \times 1$; the CIFAR-10 dataset, composed of 50000 training and 10000 test images with dimensionality $32 \times 32 \times 3$; the STL-10 dataset, composed of 5000 training and 8000 tests images with dimensionality $96 \times 96 \times 3$ plus 100000 unlabeled images; and the SVHN dataset, composed of 604388 training and 26032 test images with dimensionality $32 \times 32 \times 3$. No preprocessing or data augmentation of any kind was performed in these datasets during training or test phases.

---

[1]A Tensorflow demo is available at [*link removed to ensure anonymity*].

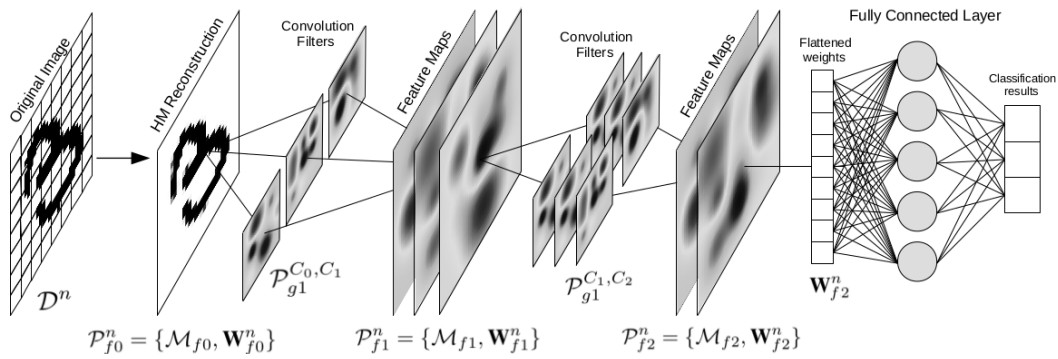

Figure 3: Diagram of a 2-layer CCNN for image classification.

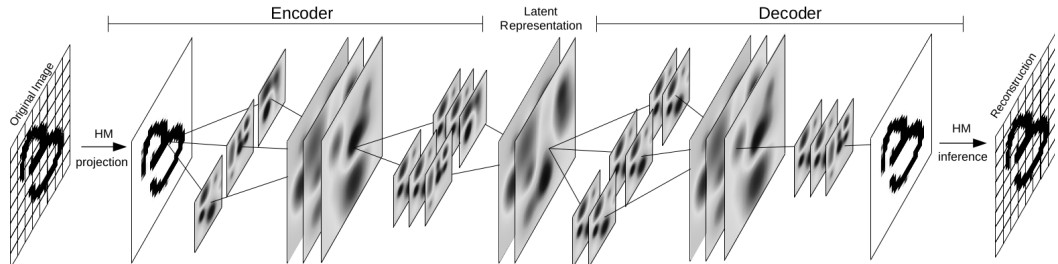

Figure 4: Diagram of a 2-layer CCAE for image reconstruction and unsupervised parameter initialization.

Examples of reconstruction results for the MNIST and CIFAR-10 datasets, using the proposed HL framework, are depicted in Figure 5. These results were obtained by projecting all images from each dataset into the same input RKHS, defined by the cluster set $\mathcal{M}_{f0}$, and producing the weight parameters $\mathbf{W}_{f0}^n$ individual to each image (for RGB images, each channel was treated independently). Both the cluster set parameters and individual model weights were optimized using the proposed joint learning methodology from Section 2.3, to minimize the squared reconstruction error. Once training was complete, these parameters served as input for a continuous convolutional neural network (CCNN) for image classification, in which each projected image is mapped to its corresponding label via cross-entropy minimization and a *softmax* activation function in the output layer (see Figure 3). To initialize the convolutional parameters of this network, a continuous convolutional auto-encoder (CCAE) was used, mirroring convolutional layers to produce a final reconstruction of the original input, via direct squared error minimization over the discretized output (see Figure 4).

To test the expressiveness of the proposed continuous feature maps, we compared CCNN image classification results against the standard DCNN (Discrete Convolutional Neural Network) architecture, in the special case when very few filters are used (here, ranging from 1 to 20, with size $3 \times 3$ for discrete and 9 clusters for continuous). A single convolutional layer was used, followed by a fully-connected layer with 200 nodes and the output layer (no dropout or regularization of any sort was used). Classification results for the MNIST dataset are depicted in Figure 6, where we can see that a single continuous filter is able to achieve better overall results than all twenty discrete filters, both in training and test accuracy. Interestingly, while a single discrete filter actually achieves worst loss function training values than a straightforward fully connected neural network (FCNN) without convolutional layers, a single continuous filter continues to improve the loss function over time, which was still decreasing after the alloted number of iterations. We also noticed less over-fitting,

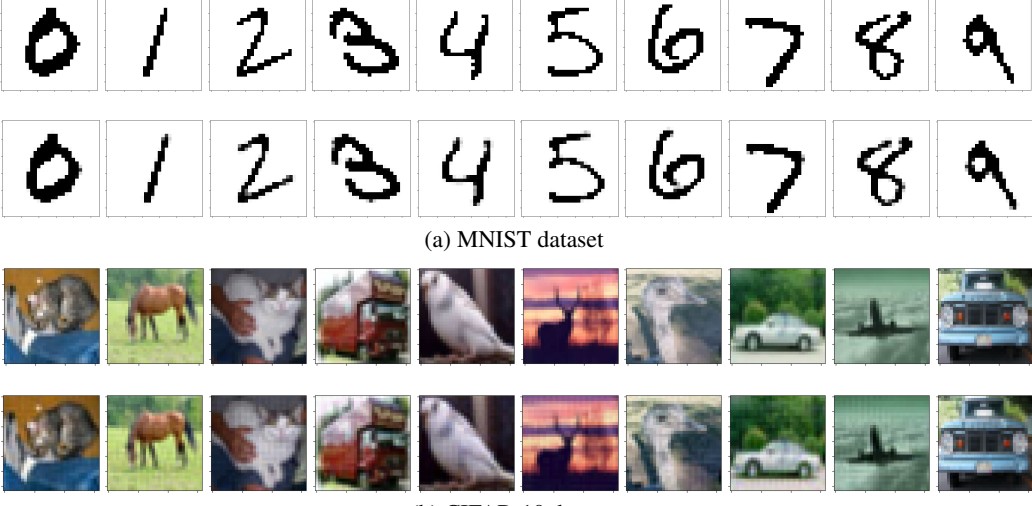

(a) MNIST dataset

(b) CIFAR-10 dataset

Figure 5: Examples of data used during experiments. The top row shows the original images, and the bottom row shows their corresponding reconstructions using the proposed HL framework.

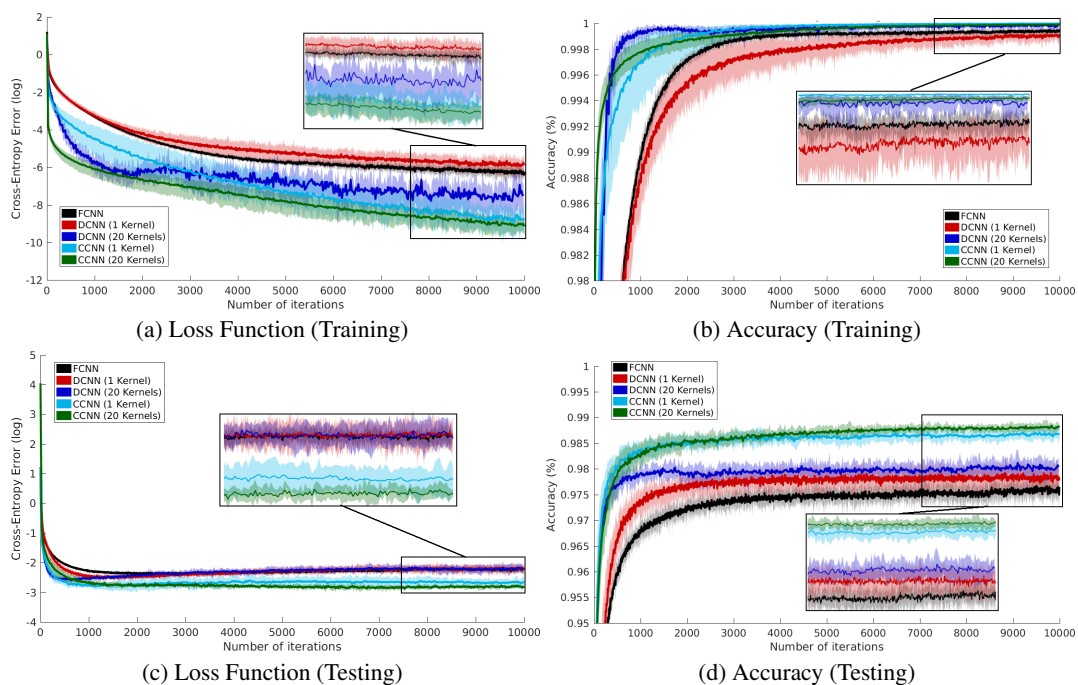

(a) Loss Function (Training)

(b) Accuracy (Training)

(c) Loss Function (Testing)

(d) Accuracy (Testing)

Figure 6: Comparison between classification results for the MNIST dataset, using different techniques. Lines indicate average values from 20 independent runs, and shades indicate min-max values between all runs.

as it can be shown by loss function values for testing data, that started to consistently increase for DCNN after a certain number of training iterations, while CCNN was able to maintain lower values throughout the entire training process. Furthermore, we can see that CCNN produces much larger ranges of loss function values both for training and testing data, indicating that the choice of initial parameter values play a more significant role during the optimization process, especially when fewer filters are considered (which is to be expected, since they are able to capture a larger range of patterns to use during the convolution process). Examples of convolutional filters obtained in these experiments are depicted in Figure 7, where we can see their variability and ability to model different patterns that will be useful during the classification process.

Classification results, in terms of percentual test accuracy error, for the three datasets considered here are presented in Tables 1-4, in relation to other image classification techniques found in the literature. The same CCNN architecture was used in all cases, composed of three convolutional layers with 20-40-60 filters of sizes 25-16-9 and pooling ratios of 2-3-4 in relation to input data dimen-

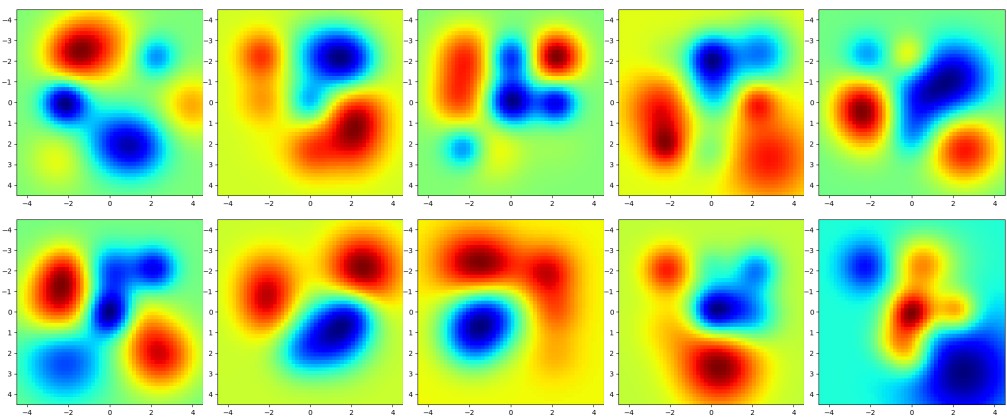

Figure 7: Examples of CCNN filters for the MNIST dataset.

| Method | Acc. |
|---|---|
| Frac. Max-Pooling (Graham, 2015) | 99.68 |
| **CCNN (CCAE init.)** | **99.63** |
| Conv. Kernel Net. (Mairal et al., 2014) | 99.61 |
| Maxout Net. (Goodfellow et al., 2013) | 99.55 |
| **CCNN (random init.)** | **99.51** |
| PCANet (Chan et al., 2014) | 99.38 |

Table 1: MNIST results.

| Method | Acc. |
|---|---|
| Frac. Max-Pooling (Graham, 2015) | 96.53 |
| Maxout Net. (Goodfellow et al., 2013) | 90.65 |
| **CCNN (CCAE init.)** | **87.48** |
| Conv. Kernel Net. (Mairal et al., 2014) | 82.18 |
| **CCNN (random init.)** | **81.95** |
| PCANet (Chan et al., 2014) | 78.67 |

Table 2: CIFAR-10 results.

| Method | Acc. |
|---|---|
| Multi-Task Bayes (Swersky et al., 2013) | 70.10 |
| C-SVDDNet (Wang et al., 2014) | 68.23 |
| **CCNN (CCAE init.)** | **63.81** |
| Conv. Kernel Net. (Mairal et al., 2014) | 62.32 |
| Disc. Learning (Gens et al., 2012) | 62.30 |
| Pooling Invariant (Jia et al., 2012) | 58.28 |

Table 3: STL-10 results.

| Method | Acc. |
|---|---|
| ReNet (Visin et al., 2015) | 97.62 |
| Maxout Net. (Goodfellow et al., 2013) | 97.53 |
| Stoch. Pooling (Zeiler et al., 2013) | 97.02 |
| **CCNN (CCAE init.)** | **96.27** |
| Shallow CNN (McDonnell et al., 2015) | 96.02 |
| **CCNN (random init.)** | **93.48** |

Table 4: SVHN results.

sionality, followed by two fully-connected layers with 512-1024 nodes and 0.5 dropout (Srivastava et al., 2014). Note that this architecture is much simpler than the ones presented by networks capable of achieving state-of-the-art classification results in these datasets, possessing a total of 144596 convolutional parameters, $588 + 261 + 147 = 996$ of which define the intermediary RKHS for feature maps representation and the remaining $6000 + 51200 + 86400 = 143600$ representing the filters within these RKHS. Nevertheless, we can see that the proposed convolutional Hilbert layer is able to achieve competitive results in all three datasets, even with such shallow and narrow architecture, which further exemplifies the descriptive power of a continuous representation when applied in conjunction with the convolution operation. Particularly, the introduction of unsupervised pre-training, using the proposed CCAE architecture to generate initial parameter estimates, significantly improves accuracy results.

## 5 CONCLUSION

This paper introduced a novel technique for data representation that takes place in a high-dimensional Reproducing Kernel Hilbert Space (RKHS), where arbitrarily complex functions can be approximated in a continuous fashion using a series of simple kernels. We show how these kernels can be efficiently convolved to produce approximations of convolution results between two functions in different RKHS, and how this can be applied in an image classification scenario, via the introduction of a novel deep learning architecture entitled Continuous Convolutional Neural Networks (CCNN). Experimental tests using standard benchmark datasets show that this proposed architecture is able to achieve competitive results with much smaller network sizes, by focusing instead on more descriptive individual filters that are used to extract more complex patterns. Although promising, there are still several potential improvements that are left for future work, such as: RKHS sparsification, so only a subset of clusters are used for feature vector calculation, which would greatly improve computational speed and memory requirements; different learning rates and optimization strategies for each class of parameter (cluster location, length-scale and weight), to improve convergence rates; and the use of different kernels for feature vector representation, as a way to encode different properties in the resulting feature maps.

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
