# OpenReview forum: "Continuous Convolutional Neural Networks for Image Classification"
_ICLR.cc/2018/Conference — Reject_

### Official Review · AnonReviewer1 · 2017-11-27
**nice idea. lack in motivation and insights**

**Rating:** 5
**Confidence:** 3

**Review:**

The paper introduces the notion of continuous convolutional neural networks.
The main idea of the paper is to project examples into an RK Hilbert space
and performs convolution and filtering into that space. Interestingly, the
filters defined in the Hilbert space  have parameters that are learnable.

While the idea may be novel and interesting, its motivation is not clear for
me. Is it for space? for speed? for expressivity of hypothesis spaces?
Most data that are available for learning are in discrete forms and hopefully,
they have been digitalized according to Shannon theory. This means that they bring
all necessary information for rebuilding their continuous counterpart. Hence, it is
not clear why projecting them back into continuous functions is of interest.

Another point that is not clear or at least misleading is the so-called Hilbert Maps.
As far as I understand, Equation (4) is not an embedding into an Hilbert space but
is more a proximity space representation [1]. Hence, the learning framework of the
authors can be casted more as a learning with similarity function than learning
into a RKHS [2]. A proper embedding would have mapped $x$ into a function
belonging to $\mH$. In addition, it seems that all computations are done
into a \ell^2 space instead of in the RKHS (equations 5 and 11).
Learning good similarity functions is also not novel [3] and Equations
(6) and (7) corresponds to learning these similarity functions.
As far as I remember, there exists also some paper from the nineties that
learn the parameters of RBF networks but unfortunately I have not been able to
google some of them.


Part 3 is the most interesting part of the paper, however it would have been
great if the authors provide other kernel functions with closed-form convolution
formula that may be relevant for learning.
The proposed methodology is evaluated on some standard benchmarks in vision. While
results are pretty good, it is not clear how the various cluster sets have been obtained
and what are their influence on the performances (if they are randomly initialized, it
would be great to see standard deviation of performances with respect to initializations).
I would also be great to have intuitions on why a single continuous filter works betters
than 20 discrete ones (if this behaviour is consistent accross initialization).

On the overall, while the idea may be of interested, the paper lacks in motivations
in connecting to relevant previous works and in providing insights on why it works.
However, performance results seem to be competitive and that's the reader may
be eager for insights.


minor comments
---------------

* the paper employs vocabulary that is not common in ML. eg. I am not sure what
occupancy values, or inducing points are.

* Supposingly that the authors properly consider computation in RKHS, then \Sigma_i
should be definite positive right? how update in (7) is guaranteed to be DP?
This constraints may not be necessary if instead they used proximity space representation.





[1] https://alex.smola.org/papers/1999/GraHerSchSmo99.pdf
[2] https://www.cs.cmu.edu/~avrim/Papers/similarity-bbs.pdf
[3] A. Bellet, A. Habrard and M. Sebban. Similarity Learning for Provably Accurate Sparse Linear Classification.

---

> ### Author Response · Authors · 2017-12-25
> **Rebuttal for Reviewer 3 (Part 2)**
>
> ----------
> – Q: The proposed methodology is evaluated on some standard benchmarks in vision. While results are pretty good, it is not clear how the various cluster sets have been obtained and what are their influence on the performances (if they are randomly initialized, it would be great to see standard deviation of performances with respect to initializations).
> – A: Although not required, in all experiments the cluster set was initialized as a grid, with mean values equally spaced and the same variance value, so that the distance between mean values is equal to two standard deviations (weight values are initialized randomly, using a Gaussian distribution with mean 0 and variance 0.1). This was done to guarantee a good coverage of the entire input space even with a small number of clusters. These values were then optimized accordingly (input data using the joint kernel learning methodology from Section 2.3 and feature maps using the classification methodology from Section 3). This was clarified on the paper, to facilitate the reader’s understanding.
> ----------
> – Q: I would also be great to have intuitions on why a single continuous filter works betters than 20 discrete ones (if this behaviour is consistent accross initialization). On the overall, while the idea may be of interested, the paper lacks in motivations in connecting to relevant previous works and in providing insights on why it works. However, performance results seem to be competitive and that's the reader may be eager for insights.
> – A: Projecting discrete data into a continuous function in a RKHS provides an alternative method of data representation, which we can exploit to produce more descriptive feature maps. For example, we are not constrained to a fixed-size grid map, but rather have inducing points that are free to move around the input space, and these positions, alongside other kernel parameters (i.e. variance values) are learned during the training process in conjunction with the more traditional weight values. This produces certain degrees of freedom in the learning process that cannot be achieved with standard discrete convolutional kernels, especially when dealing with such narrow and shallow topologies. We provide connections with previous works on Hilbert Maps, and with tangentially similar works on RKHS projection for convolution, however the proposed methodology is novel and still has not been explored in a neural network context, for deep learning purposes.
> ----------
> – Q: The paper employs vocabulary that is not common in ML. eg. I am not sure what occupancy values, or inducing points are.
> – A: Occupancy values are simply the probability of a given input point to be occupied or not, varying from 0.0 (not occupied) to 0.5 (unknown) and 1.0 (occupied). They are given by the classifier used as the occupancy model, based on input points projected into the RKHS. The inducing set is used to approximate training data using a smaller subset of points, for computational purposes (the number M of inducing points is much smaller than the number N of training points, M << N). Once optimization is complete, the training data can be discarded and only the inducing set is maintained, which greatly decreases memory requirements. These terms were clarified in the paper, to facilitate the reader’s understanding.
> ----------
> – Q: Supposing that the authors properly consider computation in RKHS, then \Sigma_i should be definite positive right? how update in (7) is guaranteed to be DP? This constraints may not be necessary if instead they used proximity space representation.
> – A: To guarantee positive-definiteness, we are in fact learning a lower triangular matrix V, which is then used to produce  \Sigma_i = U^T . U, a positive-definite matrix.  U is assumed to be invertible (i.e. it has no zeros on the main diagonal), which indeed cannot be guaranteed during the optimization process, however that was never the case in any of the experiments. We attribute this behavior to the initialization procedure, which places a large variance value for each kernel, so it stabilizes before reaching values close to zero (the noisy nature of stochastic gradient descent also naturally avoids exact values of zero for trainable parameters). This has been clarified on the paper, to facilitate the reader’s understanding.

---

> ### Author Response · Authors · 2017-12-25
> **Rebuttal for Reviewer 3 (Part 1)**
>
> ----------
> – Q: While the idea may be novel and interesting, its motivation is not clear for me. Is it for space? for speed? for expressivity of hypothesis spaces? Most data that are available for learning are in discrete forms and hopefully, they have been digitalized according to Shannon theory. This means that they bring all necessary information for rebuilding their continuous counterpart. Hence, it is not clear why projecting them back into continuous functions is of interest.
> – A: We have shown in the paper that continuous kernels produce much more descriptive convolutional filters, and thus can achieve similar classification results with much smaller network sizes. This translates into smaller memory requirements, faster computational speeds and higher expressivity, all very attractive properties in Deep Learning applications. Furthermore, projecting information as continuous functions in the RKHS produces an alternative form of data representation, that can be exploited for learning purposes. For example, spatial dependencies are modeled using kernels functions and thus are not constrained to grid representations, producing properties such as adaptive support, non-stationarity and automatic relevance determination. We are not arguing that our approach is better or worse than standard discrete convolution, but rather that it is different and worth exploring by the scientific community. Also, because it introduces a novel data representation for learning purposes, ICLR would be the perfect vehicle for a first submission.
> ----------
> – Q: Another point that is not clear or at least misleading is the so-called Hilbert Maps. As far as I understand, Equation (4) is not an embedding into an Hilbert space but is more a proximity space representation [1]. Hence, the learning framework of the authors can be casted more as a learning with similarity function than learning into a RKHS [2]. A proper embedding would have mapped $x$ into a function belonging to $\mH$. In addition, it seems that all computations are done into a \ell^2 space instead of in the RKHS (equations 5 and 11). Learning good similarity functions is also not novel [3] and Equations (6) and (7) corresponds to learning these similarity functions. As far as I remember, there exists also some paper from the nineties that learn the parameters of RBF networks but unfortunately I have not been able to google some of them.
> – A: Equation (4) was introduced in the original Hilbert Maps paper [7] as one of three possible feature vectors, alongside Random Fourier and Nyström features; and further explored in [8] to include non-stationarity and inducing point placement using clustering. Further theoretical details on how this is an embedding into a Hilbert space were left out for conciseness, since the original papers are cited accordingly. While similarity functions try to approximate input values in a high-dimensional space, for tasks such as clustering, a feature vector is used to approximate popular kernels using dot products (Section 2.2 was rewritten to clarify this property). Equations (5) and (11) use block-matrices K containing, in each row, the feature vector corresponding to each input point, which is then multiplied by the weights to produce a single scalar for each row, which is then added to the bias. This is standard procedure for the Hilbert Maps training and inference process, but rewritten as a neural network layer to allow back-propagation and joint kernel parameter learning during the training process. The parameters that perform this projection into a RKHS (i.e. mean and variance) are learned using the proposed framework, and Equations (6) and (7) provide the gradients for mean and variance values, respectively, to be used during back-propagation.
> ----------
> – Q: Part 3 is the most interesting part of the paper, however it would have been great if the authors provide other kernel functions with closed-form convolution formula that may be relevant for learning.
> – A: Although a more detailed study of different kernel functions for continuous convolution would be interesting, it does not add to the core theoretical aspects of the proposed framework, so it was left out to produce a more concise text. We are currently working on a survey of such kernels, describing their advantages and shortcomings for different learning tasks, and will include it as an appendix on the final version of the paper.

---

### Official Review · AnonReviewer2 · 2017-11-28
**This paper proposes an extension of CNN to the case where data features are continuous functions in a RKHS. The paper is well written and provides some new insights on incorporating kernels in CNN.**

**Rating:** 6
**Confidence:** 2

**Review:**

This paper aims to provide a continuous variant of CNN. The main idea is to apply CNN on Hilbert maps of the data. The data is mapped to a continuous Hilbert space via a reproducing kernel and a convolution layer is defined using the kernel matrix. A convolutional Hilbert layer algorithm is introduced and evaluated on image classification data sets.

The paper is well written and provides some new insights on incorporating kernels in CNN.

The kernel matrix in Eq. 5 is not symmetric and the kernel function in Eq. 3 is not defined over a pair of inputs. In this case, the projections of the data via the kernel are not necessarily in a RKHS. The connection between Hilbert maps and RKHS in that sense is not clear in the paper.

The size of a kernel matrix depends on the sample size. In large scale situations, working with the kernel matrix can be computational expensive. It is not clear how this issue is addressed in this paper.

In section 2.2, how \mu_i and \sigma_i are computed?

How the proposed approach can be compared to convolutional kernel networks (NIPS paper) of Mairal et al. (2014)?

---

> ### Author Response · Authors · 2017-12-25
> **Rebuttal for Reviewer 2**
>
> ----------
> – Q: The kernel matrix in Eq. 5 is not symmetric and the kernel function in Eq. 3 is not defined over a pair of inputs. In this case, the projections of the data via the kernel are not necessarily in a RKHS. The connection between Hilbert maps and RKHS in that sense is not clear in the paper.
> – A: Equation (3) is defined over pairs of inputs in the sense that they correlate input points with inducing points, according to a covariance matrix that acts as length-scale. The equation was rewritten for clarity, and a better explanation on this behavior was provided to facilitate the reader’s understanding. The kernel matrix in Equation (5) is not square, so it cannot be symmetrical (i.e. it is not a covariance matrix). It is a N x M matrix containing, in each row, the feature vector corresponding to each input point, which is then multiplied by the weights to produce a single scalar for each row, which is then added to the bias. This is standard procedure for the Hilbert Maps training and inference process, but rewritten as a neural network layer to allow back-propagation and joint kernel parameter learning during the training process.
> ----------
> – Q: The size of a kernel matrix depends on the sample size. In large scale situations, working with the kernel matrix can be computational expensive. It is not clear how this issue is addressed in this paper.
> – A: The number of inducing points used for RKHS projection is typically much smaller than the number of training points (especially at higher dimensions), which alleviates large-scale issues. Additionally, the proposed framework can be sparsified, by considering only a subset of inducing points when calculating the feature vector for each input point. This strategy has been successfully applied in a Gaussian process context [4], and can be easily extended to the proposed framework without minimal modifications. This was not addressed in this paper due to software limitations when dealing with back-propagation through sparse matrices, however as mentioned it is planned for future work and stable code release.
> ----------
> – Q: In section 2.2, how \mu_i and \sigma_i are computed?
> – A: In the original Hilbert Maps paper, the authors cluster input data and use each subset of points to calculate statistical mean and variance values. In this paper, these values are obtained using the joint kernel learning methodology proposed in Section 2.3 to produce optimal weight, mean and variance values from initial guesses. In all experiments, the clusters were initialized as a grid, with mean values equally spaced and the same variance value, so that the distance between mean values is equal to two standard deviations (weight values are initialized randomly, using a Gaussian distribution with mean 0 and variance 0.1). This has been clarified on the paper, to facilitate the reader’s understanding.
> ----------
> – Q: How the proposed approach can be compared to convolutional kernel networks (NIPS paper) of Mairal et al. (2014)?
> – A: To the best of our knowledge, the works of [5,6] are the most similar to the proposed methodology, in the sense that both apply RKHS projections using kernels to produce convolutional results in a a multi-layer neural network. However, there are key differences in how this is achieved, most notably because Convolutional Kernel Networks (CKN) still rely on discrete image patches, that are projected individually into the RKHS via the kernel function, and its parameters are the same as in standard discrete convolution (number of layers, number of filters, shape of filters and size of feature maps), while the others (\beta_k and \sigma_k) are automatically chosen. On the other hand, the proposed methodology first projects the entire input data into the RKHS via the kernel functions, and then performs convolution directly in this projected continuous function, without ever touching the original dataset again. Additionally, the proposed methodology also learns extra kernel parameters (i.e. mean and variance) on top of the standard discrete convolution parameters. This analysis has been added to the paper, for a better understanding of the differences between these two techniques.

---

### Official Review · AnonReviewer4 · 2017-12-20
**review: experimental results not compelling**

**Rating:** 4
**Confidence:** 4

**Review:**

This paper formulates a variant of convolutional neural networks which models both activations and filters as continuous functions composed from kernel bases. A closed-form representation for convolution of such functions is used to compute in a manner than maintains continuous representations, without making discrete approximations as in standard CNNs.

The proposed continuous convolutional neural networks (CCNNs) project input data into a RKHS with a Gaussian kernel function evaluated at a set of inducing points; the parameters defining the inducing points are optimized via backprop. Filters in convolutional layers are represented in a similar manner, yielding a closed-form expression for convolution between input and filters. Experiments train CCNNs on several standard small-scale image classification datasets: MNIST, CIFAR-10, STL-10, and SVHN.

While the idea is interesting and might be a good alternative to standard CNNs, the paper falls short in terms of providing experimental validation that would demonstrate the latter point. It unfortunately only experiments with CCNN architectures with a small number (eg 3) layers. They do well on MNIST, but MNIST performance is hardly informative as many supervised techniques achieve near perfect results. The CIFAR-10, STL-10, and SVHN results are disappointing. CCNNs do not outperform the prior CNN results listed in Table 2,3,4. Moreover, these tables do not even cite more recent higher-performing CNNs. See results table in (*) for CIFAR-10 and SVHN results on recent ResNet and DenseNet CNN designs which far outperform the methods listed in this paper.

The problem appears to be that CCNNs are not tested in a regime competitive with the state-of-the-art CNNs on the datasets used. Why not? To be competitive, deeper CCNNs would likely need to be trained. I would like to see results for CCNNs with many layers (eg 16+ layers) rather than just 3 layers. Do such CCNNs achieve performance compatible with ResNet/DenseNet on CIFAR or SVHN? Given that CIFAR and SVHN are relatively small datasets, training and testing larger networks on them should not be computationally prohibitive.

In addition, for such experiments, a clear report of parameters and FLOPs for each network should be included in the results table. This would assist in understanding tradeoffs in the design space.

Additional questions:

What is the receptive field of the CCNNs vs those of the standard CNNs to which they are compared? If the CCNNs have effectively larger receptive field, does this create a cost in FLOPs compared to standard CNNs?

For CCNNs, why does the CCAE initialization appear to be essential to achieving high performance on CIFAR-10 and SVHN? Standard CNNs, trained on supervised image classification tasks do not appear to be dependent on initialization schemes that do unsupervised pre-training. Such dependence for CCNNs appears to be a weakness in comparison.

---

> ### Author Response · Authors · 2017-12-25
> **Rebuttal for Reviewer 1**
>
> ----------
> – Q: Experiments train CCNNs on several standard small-scale image classification datasets: MNIST, CIFAR-10, STL-10, and SVHN. While the idea is interesting and might be a good alternative to standard CNNs, the paper falls short in terms of providing experimental validation that would demonstrate the latter point. It unfortunately only experiments with CCNN architectures with a small number (eg 3) layers. CCNNs do not outperform the prior CNN results listed in Table 2,3,4. Moreover, these tables do not even cite more recent higher-performing CNNs. The problem appears to be that CCNNs are not tested in a regime competitive with the state-of-the-art CNNs on the datasets used. Why not?
> – A: We agree that experimental results are not on par with the latest achievements in these datasets, however we would like to point out that the CCNN topologies used in this paper are much simpler than standard CNN state-of-the-art counterparts, containing only a fraction of the number of trainable parameters, and do not include many of the regularization techniques and optimization tricks commonly used to avoid these shortcomings. This was a choice, so we can analyze this novel technique by itself in a more pure state, without relying on quick fixes that are already available in the literature and can be easily incorporated regardless of which convolutional layer (continuous or discrete) is utilized. Additionally, the proposed framework consistently outperforms Convolutional Kernel Networks [6], which is currently the most well-known deep learning approach that relies on kernel functions and RKHS projections. Stable code will be released with the paper, and we will encourage and work alongside interested parties in order to test the proposed framework under different conditions, but we believe a first submission should focus more on the theoretical aspects and less on fine-tuning for optimal performance. And, as a conference on learning representations, ICLR would be the perfect vehicle to introduce a novel methodology for data modeling in deep learning tasks.
> ----------
> – Q: What is the receptive field of the CCNNs vs those of the standard CNNs to which they are compared? If the CCNNs have effectively larger receptive field, does this create a cost in FLOPs compared to standard CNNs?
> – A: The proposed framework does not have a fixed receptive field, but rather a fixed number of inducing points that compose each feature map. The location (and variance) of these inducing points is optimized during training, so they can be further or nearer the center of the feature map as needed, in order to minimize the cost function. Therefore, a CCNN can have a larger receptive field in comparison to a CNN without necessarily increasing FLOPs. The number of inducing points for the proposed classification topology is described in the experiments section, and vary for each layer of the neural network (25-16-9). If converted to receptive field sizes, these are within the standard sizes for CNNs (5x5, 4x4 and 3x3).
> ----------
> – Q: For CCNNs, why does the CCAE initialization appear to be essential to achieving high performance on CIFAR-10 and SVHN? Standard CNNs, trained on supervised image classification tasks do not appear to be dependent on initialization schemes that do unsupervised pre-training. Such dependence for CCNNs appears to be a weakness in comparison.
> – A: The convolutional filters in a CCNN are more expressive than in a standard CNN, and therefore have more degrees of freedom, which creates more stable suboptimal solutions during the optimization process. The CCAE initialization provides better starting points for these convolutional filters, so they can converge to more optimal solutions. We agree that this is a weakness, however it is worth mentioning that the CCNN topologies used in experiments are much simpler than standard CNN state-of-the-art counterparts, and do not include many of the regularization techniques and optimization tricks commonly used to avoid these shortcomings. This was a choice, so we can analyze this novel technique by itself in a more pure state, without relying on quick fixes that are already available in the literature and can be easily incorporated to mask otherwise interesting behaviors (such as this one).

---

### Public Comment · (anonymous) · 2017-10-22
**Related Literature on Continuous Neural Networks**

Very interesting approach to continuously relax convolutional filters. There's been substantial literature on infinite dimensional neural networks, such as your proposal, since the early 90s, that you might want to cite in your work. This paper (https://openreview.net/forum?id=H1pri9vTZ) outlines all of these approaches and gives some theoretical justification for your relaxation and others like it.


Best of luck!

---

> ### Author Response · Authors · 2018-01-03
> **Thank you for the related literature**
>
> Thank you very much for this comment, we have read the paper and incorporated some discussion of these approaches into our own paper, including how they are connected and how our proposed framework differs from previous methods.

---

### Author Response · Authors · 2017-12-25
**References for Rebuttals**

------------------
[1] https://alex.smola.org/papers/1999/GraHerSchSmo99.pdf
[2] https://www.cs.cmu.edu/~avrim/Papers/similarity-bbs.pdf
[3] A. Bellet, A. Habrard and M. Sebban. Similarity Learning for Provably Accurate Sparse Linear Classification. International Conference on Machine Learning (ICML), 2012.
[4] S. Vasudevan, F. Ramos, E. Nettleton, and H. Durrant-Whyte. Gaussian process modeling of large-scale terrain. Journal of Field Robotics (JFR), 26(10):812–840, 2010.
[5] J. Mairal, P. Koniusz, Z. Harchaoui, and C. Schmid. Convolutional kernel networks. In
arXiv:1406.3332, 2014.
[6] J. Mairal. End-to-end kernel learning with supervised convolutional neural networks. In Advances in Neural Information Processing Systems (NIPS), 2016.
[7] F. Ramos and L. Ott. Hilbert maps: Scalable continuous occupancy mapping with stochastic gradient descent. In Proceedings of Robotics: Science and Systems (RSS), 2015.
[8] V. Guizilini and F. Ramos. Large-scale 3d scene reconstruction with Hilbert maps. In Proceedings of the IEEE International Conference on Intelligent Robots and Systems (IROS), 2016.

---

### Decision · Program_Chairs · 2018-01-29
**ICLR 2018 Conference Acceptance Decision**

**Decision:**

Reject

**Comment:**

The paper received borderline negative scores: 5,6,4.

The authors response to R1 question about the motivations was "...thus can achieve similar classification results with much smaller network sizes. This translates into smaller memory requirements, faster computational speeds and higher expressivity." If this is really the case, then some experimental comparison to compression methods (e.g. Song Han's PhD work at Stanford) is needed to back up this.

R4 raises issues with the experimental evaluation and the AC agrees with them that they are disappointing. In general R4 makes some good suggestions for improving the paper.

The author's rebuttal also makes the general point that the paper should be accepted as it contains ideas, that these are sufficient alone: "We strongly believe that with some fine-tuning it could achieve considerably better results, however we also believe that this is not the point in a first submission...". The AC disagrees with this. Ideas are cheap. *Good ideas*, i.e. those that work, as in get good performance on standard benchmarks are valuable however. The reason for having benchmarks is to give some of objective way of seeing if an idea has any merit to it. So while the reviewers and the AC accept that the paper has some interesting ideas, this is not enough for warrant acceptance.